# A One-Year Longitudinal Study: Changes in Depression and Anxiety in Frontline Emergency Department Healthcare Workers in the COVID-19 Pandemic

**DOI:** 10.3390/ijerph182111228

**Published:** 2021-10-26

**Authors:** Francesca Th’ng, Kailing Adriel Rao, Lixia Ge, Desmond Mao, Hwee Nah Neo, Joseph Antonio De Molina, Eillyne Seow

**Affiliations:** 1Acute & Emergency Care Department, Khoo Teck Puat Hospital, 90 Yishun Central, Singapore 768828, Singapore; rao.adriel.k@ktph.com.sg (K.A.R.); mao.desmond.r@ktph.com.sg (D.M.); neo.hwee.nah@ktph.com.sg (H.N.N.); seow.eillyne@ktph.com.sg (E.S.); 2Health Services and Outcomes Research, National Healthcare Group, 3 Fusionopolis Link, Singapore 138543, Singapore; Lixia_GE@nhg.com.sg (L.G.); Joseph_Antonio_MOLINA@nhg.com.sg (J.A.D.M.)

**Keywords:** COVID-19, depression, anxiety, emergency department, healthcare workers, mental health

## Abstract

Frontline healthcare workers (HCWs) fighting COVID-19 have been associated with depression and anxiety, but there is limited data to illustrate these changes over time. We aim to quantify the changes in depression and anxiety amongst Emergency Department (ED) HCWs over one year and examine the factors associated with these changes. In this longitudinal single-centre study in Singapore, all ED HCWs were prospectively recruited face-to-face. Paper-based surveys were administered in June 2020 and June 2021. Depression and anxiety were measured using DASS-21. The results of 241 HCWs who had completed both surveys were matched. There was significant improvement in anxiety amongst all HCWs (Mean: 2020: 2.85 (±3.19) vs. 2021: 2.54 (±3.11); Median: 2020: 2 (0–4) vs. 2021: 2 (0–4), *p* = 0.045). HCWs living with elderly and with concerns about infection risk had higher odds of anxiety; those living with young children had lower odds of anxiety. There was significant worsening depression amongst doctors (Mean: 2020: 2.71 (±4.18) vs. 2021: 3.60 (±4.50); Median: 2020: 1 (0–3) vs. 2021: 3 (0–5), *p* = 0.018). HCWs ≥ 41 years, living with elderly and with greater concerns about workload had higher odds of depression. HCWs who perceived better workplace support and better social connectedness had lower odds of depression. In summary, our study showed significant improvement in anxiety amongst ED HCWs and significant worsening depression amongst ED doctors over one year. Age, living with elderly, and concerns about workload and infection risk were associated with higher odds of depression and anxiety.

## 1. Introduction

The effects of the COVID-19 pandemic on the mental health outcomes (MHOs) of healthcare workers (HCWs) worldwide are well-documented. Since the start of the COVID-19 pandemic more than a year ago, healthcare systems around the world continue to experience high levels of demand on all resources. With the emergence of highly resistant strains and multiple waves of infection in various countries, frontline HCWs have continued to risk infection, endure increased workloads, suffer trauma by witnessing first-hand high levels of death and morbidity, as well as deal with challenges arising from resource allocation. A large meta-analysis by Batra et al., consisting of 65 studies carried out amongst HCWs, showed that the pooled prevalence for anxiety was 34.4%, depression 31.8%, stress 40.3%, post-traumatic stress syndrome 11.4%, insomnia 27.8%, psychological distress 46.1% and burnout 37.4% [1]. There was higher prevalence of depression and anxiety among females, nurses and frontline workers. A study carried out in UK and USA found that frontline HCWs had at least a threefold increased risk of reporting a positive test for COVID-19 compared with the general community [2]. In addition to what has been mentioned in the meta-analysis, poorer MHOs amongst frontline HCWs have also been associated with those living alone and worries about being infected or infecting others [3,4,5,6,7,8]. 

There are currently limited published longitudinal studies amongst frontline HCWs to quantify the ongoing psychological impact of COVID-19 even as the pandemic situation waxes and wanes. Two large prospective longitudinal studies amongst Chinese frontline HCWs showed slightly different outcomes: one study [9] found significantly worse psychiatric status (somatization, obsessive-compulsiveness, interpersonal sensitivity, depression, anxiety, hostility, phobic anxiety, paranoid ideation, psychoticism) and sleep quality a month later; and the other [5] found significantly higher risks for depression, anxiety and PTSD symptoms during the outbreak period compared to the stable period of the pandemic a month later. On a smaller scale, a study in Belgium [10] amongst Intensive Care Unit (ICU) nurses in April 2020 showed that they had improved depression, anxiety and somatization over a two-month period. Similarly, a study done in Singapore [6] in March 2020 amongst residents in training showed that residents who were deployed to the higher risk National Centre for Infectious Diseases (NCID) to manage patients with COVID-19 had lower perceived stress and stigma at the three-month follow-up. To date, there have been no published longitudinal studies to assess the changes in depression and anxiety in individual frontline Emergency Department (ED) HCWs over a more prolonged period, other than the time frames in the above-mentioned studies. 

In general, longitudinal studies are important to detect changes in the characteristics of the sample population, on a group and individual level. Longitudinal studies conducted in military personnel with prolonged combat exposure had shown wide-ranging adverse effects on health behaviours and mental health [11]. In the healthcare context, the effects of fighting a prolonged pandemic that has spanned longer than a year are unknown. Our previous cross-sectional study [12] among ED HCWs in June 2020 showed that 27.5% of HCWs screened positive for depression and 34.3% for anxiety. Females were more likely to have anxiety, and those living with elderly had significantly higher median anxiety scores. 

This longitudinal study is to help us to identify the factors associated with poorer MHOs in our cohort of frontline ED HCWs a year since the first wave of the pandemic. This will allow us to optimize the support for their wellbeing during their fight in this protracted pandemic. Resilience is the ability to adapt to adversity or stressful situations. During a crisis like the COVID-19 outbreak, it has been shown that enhancing resilience and coping strategies has important effects on enhancing mental health outcomes. Predictors of good resilience include having pursued hobbies and a positive family environment [13], and knowing the relationship between these coping strategies and MHOs would be useful. As of the time of the second survey in June 2021, the COVID-19 pandemic in Singapore has been more controlled compared to its peak in June 2020, as quantified by the reduced daily COVID-19 cases and hospitalisations. We aimed to (1) quantify the changes in depression and anxiety in our cohort of frontline ED HCWs over the year; and (2) identify factors associated with depression and anxiety. We hypothesise that depression and anxiety amongst individual HCWs will generally improve with the pandemic situation being more controlled compared to its peak last year.

## 2. Materials and Methods

### 2.1. Study Design and Participants

This is a longitudinal single-centre study carried out on ED HCWs in Khoo Teck Puat Hospital (KTPH), Singapore. The study hospital is a 795-bed acute hospital, and the ED sees an average of 135,000 patients a year. To date (July 2021), the study ED has seen 1635 COVID-19 positive patients.

The data used for this study were collected from two waves of surveys conducted amongst the ED HCWs in KTPH. The first wave was conducted from 1 to 9 June 2020 [12] and the second wave was conducted one year after from 1 to 16 June 2021. The methodology for the first wave survey has been described previously [12]. 

The inclusion criteria for this study were all HCWs from this single-centre’s ED. HCWs who had resigned or had transferred out of the department during the 1 year were excluded from the second survey. As participation was voluntary, HCWs who did not wish to participate were not included in the survey.

Out of the 327 ED HCWs who responded to the first survey in 2020, 19 had resigned and 22 who were temporarily deployed to ED had left ED before the second survey was conducted. These 41 HCWs were excluded from the second survey. As such, 286 ED HCWs were invited to take part in the second wave survey from 1 to 16 June 2021. Written consent was obtained from each participant. Paper-based consent forms and survey questionnaires were handed out to the eligible ED HCWs during roll calls. The ED register list was used to ensure that surveys were only handed out to each participant once. Those who were willing to take part were instructed to return the signed consent form and completed questionnaire to the investigators, either by handing in at the end of their work shift or by dropping it off directly into a collection box at the ED office. The ethic review of this study was approved by the National Healthcare Group Domain Specific Review Board (Reference number: 2020/00653 and 2021/00336).

Out of the 286 eligible ED HCWs, 279 completed the second wave survey (response rate: 97.6%). See Figure 1. Responses from the first and second waves of survey were anonymised, but they were matched based on the last four digits of the HCW’s handphone number, gender, ethnicity, and profession. In total, 241 participants were matched and their responses to both the first and second waves of survey were included for data analysis. 

There has been no specific intervention implemented by the study group during and between the two surveys. Changes of COVID-19 safety measures through the one-year period included easing of national lockdown rules (e.g., being able to see family and friends from a different household) and a period of easing of infective measures at workplace (e.g., surgical masks could be worn in non-infective areas instead of N95). COVID-19 vaccination was also offered to all HCWs for free since the beginning of 2021. Table A1 illustrates the timeline of main changes through the year in Singapore. 

### 2.2. Outcome Measures

Depression and anxiety were measured using the validated Depression, Anxiety and Stress Scale (DASS-21) [14]. DASS-21 is a 21-item self-report questionnaire and each MHO domain contains 7 items. The Depression domain assesses dysphoria, hopelessness, devaluation of life, self-deprecation, lack of interest, anhedonia and inertia; and the Anxiety domain assesses autonomic arousal, skeletal musculature effects, situational anxiety and subjective experience of anxious effect. Scores for the Depression and Anxiety domains were calculated by summing the scores for the relevant items in the respective domain. Scores were then multiplied by two to categorise individual HCWs into two groups (normal vs. positive for depression or anxiety). A positive score for depression was defined as >9 and for anxiety >7.

Demographic information including age group, gender, ethnicity, occupation and living environment were included in both surveys (Figure A1). For the 2021 survey, we included a question on vaccination status of HCWs. Further questions to capture HCWs’ concerns and perceptions (Figure A1: Sections D and E of survey), using a Likert scale with 6 options (1 = Strongly Disagree to 6 = Strongly Agree), were developed based on experts’ opinions (Study team’s ED consultants, senior nurse and biostatisticians), but they were not part of a validated instrument. These questions were later categorised based on their content relevance for data analysis (Figure A2), namely, concerns and perceptions about COVID-19 infection risk, workplace support, workload, working environment and social connectedness.

### 2.3. Statistical Analysis

Descriptive analyses were conducted to show the demographic characteristics of the included study sample. Frequency and percentage were used to describe categorical variables. Mean and standard deviation (SD) or median and inter-quantile range (IQR) were used to describe continuous variables. The distribution of the severity and status of each MHO in 2020 and 2021 were compared using Fisher’s Exact tests and their median scores were compared using Mann–Whitney U tests. Fisher’s exact tests for categorical variables and independent *t*-tests or Mann–Whitney U tests for continuous variables were performed to determine differences in demographic characteristics, as well as individual items regarding concerns and perceptions about infection, working environment, workplace support and workload and their average scores by status of each MHO.

Fixed-effects and random-effects logistic regressions on panel data were performed to identify the potential factors that were associated with individual MHOs. In each model, the status of each MHO (binary variable) was the dependent variable. The factors that were identified to be associated with any MHO (*p* < 0.1) in bivariate analysis, namely demographic characteristics, individual coping items (binary), domain scores of concerns about infection, working environment and workload, perceptions about workplace support and social connectedness, were included as the independent variables. Odds ratios (OR) and 95% confidence intervals (CIs) were reported. Hausman tests were conducted to examine whether fixed-effects or random-effects model was more appropriate. If *p* < 0.05 for the Hausman test, fixed-effects models will be used, otherwise, random-effects models will be chosen. All analyses were performed using Stata/SE 16.1. *p* < 0.05 was set as the level of significance.

## 3. Results

### 3.1. Characteristics of Study Sample

Amongst staff eligible to participate, the overall response rate for both surveys was 93.4% (Figure 1). Table 1 shows the demographic characteristics of the 241 matched participants. The majority of the participants were female (71.8%), nurses (71.4%) and Chinese (38.6%). Compared to 2020, there was a significant increase in number of participants with family member(s) or friend(s) who had contracted COVID-19 in 2021 (2020: n = 20 (8.3%) vs. 2021: n = 39 (16.2%); *p* = 0.008). As vaccination among HCWs in our hospital started in January 2021, 91.3% had received at least one dose of the COVID-19 vaccine at the time of the second survey.

Compared to the first survey, HCWs reported significantly fewer concerns about infection risk (Mean (±SD): 2020: 4.18 (±0.80) vs. 2021: 3.92 (±0.85)) and fewer concerns about their working environment in 2021 (2020: 4.14 (±0.84) vs. 2021: 3.94 (±0.99)). In 2021, HCWs also reported significantly greater concerns about workload (2020: 4.06 (±0.98) vs. 2021: 4.37 (±0.92)), lesser social connectedness (2020: 4.55 (±0.65) vs. 2021: 4.40 (±0.69)) and had perceived lesser workplace support (2020: 4.83 (±0.67) vs. 2021: 4.63 (±0.72)). See Table 1. 

### 3.2. Mental Health Outcomes

#### 3.2.1. Depression

There was no significant difference in the distribution of severity of depression between both years (Table 2). A total of 25.3% of HCWs screened positive for depression in 2020, and 28.6% in 2021 (*p* = 0.412). There was significant worsening in depression scores amongst doctors in 2021 (Mean (±SD): 2020: 2.71 ±4.18 vs. 2021: 3.60 ±4.50; Median (IQR): 2020: 1 (0–3) vs. 2021: 3 (0–5), *p* = 0.018) (Table 3). 

HCWs aged 41 years or older (Odds Ratio: 7.9, 95% CI: 1.1–55.6), those living with elderly (OR: 6.3, 95% CI: 1.8–21.8) and those with greater concerns about workload (OR: 2.0, 95% CI: 1.2–3.4) had significantly higher odds of developing depression (Table 4). The questions in the workload category that were significantly associated with depression were “There is a lack of manpower in the fever area” (*p* = 0.002) and “I spend longer hours at work since the outbreak started” (*p* = 0.010) (Table A2). HCWs who perceived better workplace support (OR: 0.5, 95% CI: 0.2–0.9) and social connectedness (OR: 0.3, 95% CI: 0.1–0.6) had significantly lower odds of developing depression. 

#### 3.2.2. Anxiety

There was no significant difference in the distribution of severity of anxiety between both years (Table 2). A total of 30.7% of HCWs were screened positive for anxiety in 2020, and 27.0% in 2021 (*p* = 0.366). There was a significant improvement in anxiety scores in our study cohort (Mean: 2020: 2.85 ± 3.19 vs. 2021: 2.54 ± 3.11; Median: 2020: 2 (0–4) vs. 2021: 2 (0–4), *p* = 0.045) (Table 3). 

HCWs living with elderly (OR: 7.9, 95% CI: 2.3–27.2) and those with greater concerns about COVID-19 infection risk (OR: 1.8, 95% CI: 1.0–3.0) had significantly higher odds of developing anxiety; those living with young children (OR: 0.1, 95% CI: 0–0.6) had significantly lower odds of developing anxiety (Table 4). 

Eighty-six HCWs were excluded from the 2020 cohort for data analysis: 19 ED staff had resigned, 22 were temporarily deployed to our ED and had left, and 45 did not match to the 2021 responses (Figure 1). This excluded group had a significantly higher percentage of anxiety (Excluded: 44.2% vs. Included: 30.7%, *p* = 0.024) and significantly higher anxiety scores (Excluded: 3 (1–6) vs. Included: 2 (0–4), *p* = 0.038) compared to the HCWs included in the 2020 cohort (Table A3).

## 4. Discussion

Our study found (1) a significant improvement in anxiety scores amongst all ED HCWs and (2) a significant worsening in depression scores amongst ED doctors over a one-year period. Significantly higher odds of depression and anxiety were associated with HCWs who were ≥41 years old, living with elderly, having concerns about infection risk and their workload. Significantly lower odds of depression and anxiety were associated with HCWs living with young children and those with better perception about workplace support and social connectedness.

It has been reported that at least one in every five HCWs suffers from depression and/or anxiety [15]. Our study, in line with a longitudinal analysis study carried out in Japan [16], showed that poor MHOs amongst HCWs were generally sustained during and between repeated outbreaks. Possible reasons for this could be that HCWs have to remain on guard against unexpected contacts, face daily infection risk whilst at work, incur transmission risk to their loved ones and increased workload [17,18,19]. 

Interestingly, another population group that has demonstrated similar poor MHOs as HCWs during this COVID-19 pandemic are college students. A large meta-analysis [20] consisting of 27 studies with 90,879 college students indicated a prevalence of 31.2% in depression and 39.4% in anxiety, with females having higher depression and anxiety than males. 

### 4.1. Overall Reduced Anxiety in HCWs 

Similar with other COVID-19 studies, including that on Belgian ICU nurses [10] over a two-month period and on Chinese HCWs over a one-month period [21], anxiety levels in HCWs, even though still high, showed a declining trend over time during a pandemic. However, evidence for this downtrend has been inconsistent [5,9,22]. One of the reasons for our higher baseline anxiety level could be that our 2020 survey was carried out just as we were exiting a ‘circuit breaker’ (national lock-down), causing anxiety scores to be high. This increased anxiety pattern during a quarantine was further demonstrated by a longitudinal study [23] on an Italian population at three time points over a month. The drop in median anxiety score in 2021 could be further explained by the fact that the excluded group of 86 HCWs in the 2020 cohort had worse anxiety. Within this excluded group, there were ED staff who had resigned and returned home to their families overseas (in our 2020 survey [12], HCWs with family overseas had poorer MHOs). Moreover, within the excluded group with worse anxiety in 2020 were HCWs temporarily deployed to our ED, and they may not have been familiar with the ED protocols and infective measures at that time. 

The odds of developing anxiety amongst our HCWs were significantly lower in those living with young children, which is contrary to other studies’ findings [24,25]. This is an interesting finding as just prior to this study, there was a surge in other COVID-19 variants and infection risk amongst children had started to climb. A possible hypothesis would be at the time of the survey, case rates in children in our country were extremely low, and we had no fatalities in children. 

Overall, the reduction in anxiety score in 2021 could be that HCWs now have better understanding of the pathophysiology and mode of transmission of the virus, that they may have familiarised themselves with the infective measure routines at work, e.g., wearing of PPE and PAPR, and have adapted socially with wearing surgical masks in public areas and social distancing [6]. Furthermore, HCWs may have higher confidence in the government over the year—clear national guidelines and infective measures had been issued, the government’s efficiency in ringfencing intermittent smaller outbreaks, the rapid roll-out of vaccination programmes including for teenagers more than 12 years of age and our relatively low daily case numbers and deaths compared to other countries. 

### 4.2. Increased Depression in Doctors

In the majority of other COVID-19 studies, including our first survey in 2020 [12], nurses were shown to have worse depression compared to doctors [3,26,27,28,29]. Reasons cited have been that nurses have closer more frequent contact with patients and worked longer hours than usual [26,27,29]. Interestingly, our cohort of matched doctors had significantly worse depression in 2021 compared to 2020, and their overall depression score in 2021 became worse than our nursing staff’s. One study which had similar outcome to ours was a Chinese cross-sectional study [17] carried out in 4 hospitals which showed that doctors had higher odds of developing moderate or severe depression (AOR 2.11 (0.96–4.64)) compared to nurses (AOR 1.66 (0.85–3.24)). Possible reasons for the worsening depression amongst our doctors could be due to the nature of their roles compared to nurses—the pressure and stress in assessing and diagnosing patients whilst being on constant guard for possible unexpected COVID-19 infection may have led to burnout [30].

Burnout is described as a state of physical, emotional and mental exhaustion that results from long-term involvement in work situations that are emotionally demanding [31]. Multiple studies have shown a reciprocal relationship between burnout and depressive symptoms. These include (1) a Portuguese cross-sectional study [32] amongst 2008 HCWs showing that higher levels of depression were significantly associated with increased levels of burnout and (2) a Finnish person-centred approach longitudinal study [33] amongst dentists over a seven-year period demonstrating that burnout and depressive symptoms clustered and developed in tandem at similar levels. It has also been widely reported that nurses experience primary and secondary traumatic stress, compassion fatigue and burnout during this COVID-19 pandemic, such as in this Chinese study [34]. However, another Spanish study [35] reported that physicians actually have higher compassion fatigue and burnout scores compared to nurses, demonstrating that both parties are likely equally vulnerable to emotional desensitization.

In a more local context, three recent Singaporean studies on HCWs also revealed that doctors were less likely to seek help for their mental health over concerns that making their struggles public may affect their licence to practice [36]. We postulate that the early signs of burnout were not well addressed in the doctor group at the start of the pandemic last year. Coupled with the inability to openly share of their struggles, these factors might have led to worsening depression scores in 2021. Because our survey was anonymous, it was easier for the doctors to report their feelings factually without the fear of being stigmatised or deemed unsafe to practice. This study’s finding of deteriorating depression amongst our doctors is crucial as this could affect our doctors’ cognitive function, precision, task performance and could have further physical ramifications. 

### 4.3. Increased Depression and Demographic Characteristics

In our current study, HCWs ≥41 years old had higher odds of developing depression. This is an interesting finding because population and healthcare studies [37,38] had shown that younger adults were associated with poorer MHOs during this COVID-19 pandemic. It was further explained that older adults’ resilience was less influenced by stressful events, that younger adults may be more worried about losing their jobs and that younger adults may be overloaded with false information on social media. In line with our results, Yildirim et al. [39] and Pan et al. [40] had shown that older HCWs were associated with poorer MHOs. This could be that older HCWs felt more vulnerable physically (in terms of infection risk, morbidity and mortality) [38], may be worried about infecting their families and may feel more exhausted with the increased workload [41]. Another vulnerable subgroup of HCWs in our study were those living with elderly family member(s)—a subgroup with poorer MHOs which has also been demonstrated in other COVID-19 studies [4]. As previously explored in our first study [12], this subgroup of HCWs may have additional stressors with caregiving itself and fear of infecting their elderly loved ones at home.

### 4.4. Increased Depression and Concerns about Workload

Our HCWs with concerns about workload, specifically with concerns about lack of manpower in the fever area and spending longer hours at work, had higher odds of developing depression. Mo et al. [42] found that increased working time per week and work intensity were risk factors for poorer MHOs. The prolonged use of PPE was also shown to result in tension, fatigue and burnout. The increased workload in our cohort could be explained by Singapore’s first hospital cluster outbreak which happened just prior to the 2021 study. That hospital was subsequently shut to ringfence the situation, and patients were re-directed to other nearby hospitals including this study’s ED. We saw a surge in our ED’s patient load, longer ED waiting times and increased bed blocks. Furthermore, to reduce infection risks between hospitals, locum HCWs who previously could move between different EDs to work could only work in one hospital, which also led to a shortage of our manpower.

### 4.5. COVID-19 Vaccination 

Despite the high COVID-19 vaccination rate (91.3%) in our 2021 cohort, there was no association between vaccination status and depression or anxiety. At the time of our study, there was no published studies looking into the association between COVID-19 vaccination and MHOs of ED HCWs. There was still no known long-term efficacy or safety profile of the vaccines, and these uncertainties could have contributed to why vaccination status had no association with improvement of MHOs. 

### 4.6. Strengths and Limitations

To our best knowledge, this is the first longitudinal study carried out in frontline ED HCWs in Asia to assess the changes in depression and anxiety during this protracted COVID-19 pandemic. This study’s longitudinal design is crucial in aiding healthcare systems to identify potential modifiable workplace factors associated with poorer MHOs over time. Furthermore, a validated assessment tool to measure depression and anxiety levels was used, and both cohorts of the study had high response rates and low numbers of missing data. The findings in our study emphasize the importance and need to develop psychological interventions to promote post-traumatic growth amongst our frontline ED HCWs. 

Limitations of this study include it being a single-centre study carried out in one department which may limit the study’s generalizability to other healthcare settings, and that socioeconomic factors, which may be confounders, had not been collected. The self-report nature of depression and anxiety under the DASS-21 scale, rather than a clinician-facilitated assessment, could also introduce bias to the results [43,44]. 

## 5. Conclusions

In summary, our longitudinal study showed that frontline ED HCWs continued to have overall poor depression and anxiety scores despite intermittent bouts of the situation being more controlled over the one-year period. There was improvement in anxiety scores amongst all ED HCWs and worsening depression scores amongst ED doctors. Factors associated with depression and anxiety were those ≥41 years old, living with elderly, with concerns about infection risk and workload. HCWs living with young children and those who perceive better workplace support and social connectedness were identified as protective factors.

The implications of this longitudinal study are that firstly, we need to recognise that effects of psychological distress inflicted by a pandemic are long-standing. From an occupational health standpoint, workplace mental health interventions cannot be sporadic, but should be regular and sustained. HCWs should continue to receive updated information about infection protocols and risk management, as well as redistribution of workload from the fever area for those most affected.

Secondly, this longitudinal study builds on the understanding that certain factors associated with depression and anxiety are non-modifiable, such as those older than 40 years of age and those living with the elderly. Gaining a better understanding of how these groups connect to fellow beings, whether through social media or lack thereof, will allow us to develop mental health programs targeted towards them. The results of the survey will also be shared with department heads and nursing leads so that they can better identify the at-risk groups to attend hospital-wide wellness initiatives. 

## Figures and Tables

**Figure 1 ijerph-18-11228-f001:**
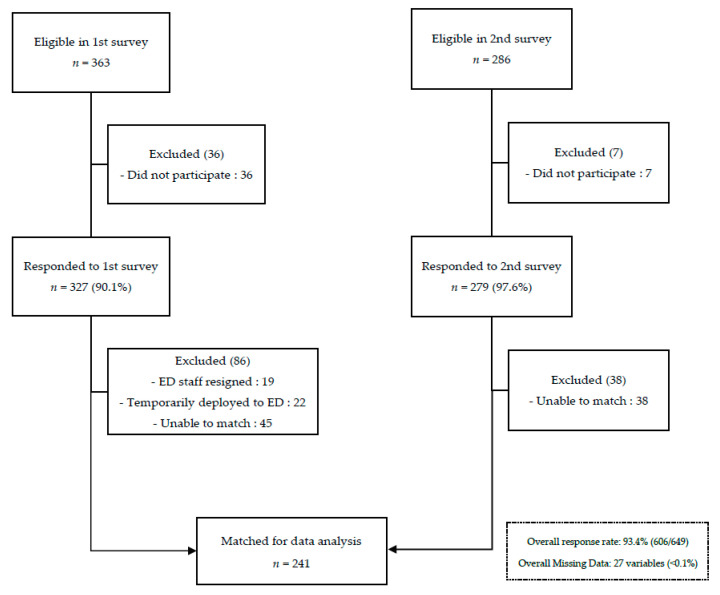
Flow diagram of the COVID-19 longitudinal study carried out in June 2020 (1st survey) and June 2021 (2nd survey).

**Table 1 ijerph-18-11228-t001:** Characteristics of matched participants (n = 241) in 2020 and 2021. There were 58 doctors (24.1%) and 183 nursing staff (75.9%) in each year’s cohort.

Characteristics	June 2020	June 2021	*p*-Value
Gender (n,%)			
Female	173 (71.8)	173 (71.8)	-
Male	68 (28.2)	68 (28.2)	
Ethnicity (n,%)			
Chinese	93 (38.6)	93 (38.6)	-
Malay	25 (10.3)	25 (10.3)	
Indian	34 (14.1)	34 (14.1)	
Filipino	77 (32.0)	77 (32.0)	
Others	12 (5.0)	12 (5.0)	
Age group in years (n,%)			
21–30	104 (43.2)	86 (35.7)	0.573
31–40	100 (41.5)	113 (46.9)	
41–50	28 (11.6)	32 (13.3)	
51–60	6 (2.5)	6 (2.5)	
>60	3 (1.2)	4 (1.7)	
Marital status (n,%)			
Single	126 (52.3)	115 (47.7)	0.605
Married	113 (46.9)	122 (50.6)	
Separated/Divorced	1 (0.4)	3 (1.2)	
Widowed	1 (0.4)	1 (0.4)	
Occupation (n,%)			
Doctors: Senior doctors	22 (9.1)	22 (9.1)	-
Junior doctors	36 (14.9)	36 (14.9)	
Nursing staff: Nurses	172 (71.4)	172 (71.4)	
Healthcare assistants	11 (4.6)	11 (4.6)	
Past medical history (n,%)			
Yes	10 (4.2)	13 (5.4)	0.522
No	231 (95.9)	228 (94.6)	
Living with young children (<12 years old) (n,%)			
Yes	36 (14.9)	49 (20.3)	0.120
No	205 (85.1)	192 (79.7)	
Living with elderly (>65 years old) (n,%)			
Yes	34 (14.1)	41 (17.0)	0.379
No	207 (85.9)	200 (83.0)	
Lives alone (n,%)			
Yes	35 (14.5)	41 (17.0)	0.453
No	206 (85.5)	200 (83.0)	
Practices a religion (n,%)			
Yes	181 (75.1)	182 (75.5)	0.916
No	60 (24.9)	59 (24.5)	
Has family or close friend with COVID-19 (n,%)			
Yes	20 (8.3)	39 (16.2)	0.008
No	221 (91.7)	202 (83.8)	
COVID-19 vaccinated (1 or 2 doses) (n,%)			
Yes	-	220 (91.3)	-
No	-	21 (8.7)	
Concerns about infection risk (Mean ± SD)	4.18 ± 0.80	3.92 ± 0.85	0.000
Concerns about working environment (Mean ± SD)	4.14 ± 0.84	3.94 ± 0.99	0.003
Concerns about workload (Mean ± SD)	4.06 ± 0.98	4.37 ± 0.92	0.000
Social connectedness (Mean ± SD)	4.55 ± 0.65	4.40 ± 0.69	0.002
Workplace support (Mean ± SD)	4.83 ± 0.67	4.63 ± 0.72	0.000

**Table 2 ijerph-18-11228-t002:** Longitudinal changes of depression and anxiety by severity in June 2020 and June 2021.

**Depression in 2020 (n)**	**Depression in 2021 (n,%)**					
	**Normal**	**Mild**	**Moderate**	**Severe**	**Extremely Severe**	**Total in 2020**
**Normal**	149 (82.3)	14 (7.8)	16 (8.9)	0	1 (0.6)	180
Positive for Depression						
Mild	15 (55.6)	6 (22.2)	3 (11.1)	0	3 (11.1)	27
Moderate	7 (29.2)	6 (24.0)	6 (24.0)	3 (12.5)	2 (8.3)	24
Severe	0	0	2 (40.0)	0	3 (60.0)	5
Extremely severe	1 (20.0)	2 (40.0)	1 (20.0)	0	1 (20.0)	5
Total in 2021	172 (71.4)	28 (11.6)	28 (11.6)	3 (1.2)	10 (4.1)	241
**Anxiety in 2020 (n)**	**Anxiety in 2021 (n,%)**					
	**Normal**	**Mild**	**Moderate**	**Severe**	**Extremely Severe**	**Total in 2020**
**Normal**	144 (86.2)	9 (5.4)	12 (7.2)	0	2 (1.2)	167
Positive for Anxiety						
Mild	16 (66.7)	1 (4.2)	5 (20.8)	1 (4.2)	1 (4.2)	24
Moderate	11 (37.9)	3 (10.3)	11 (37.9)	4 (13.8)	0	29
Severe	4 (40.0)	0	3 (30.0)	1 (10.0)	2 (20.0)	10
Extremely severe	1 (10.0)	1 (10.0)	1 (10.0)	2 (20.0)	5 (50.0)	10
Total in 2021	176 (73.0)	14 (5.8)	32 (13.3)	8 (3.3)	10 (4.1)	241

**Table 3 ijerph-18-11228-t003:** Depression and anxiety scores amongst matched HCWs (n = 241) and its subgroups of doctors (n = 58) and nursing staffs (n = 183).

MHOs	June 2020 Mean (± SD)	June 2021 Mean (± SD)	June 2020 Median (IQR)	June 2021 Median (IQR)	*p*-Value *
All HCWs (n = 241)					
Depression	3.05 ± 3.50	3.45 ± 3.91	2 (0–5)	2 (0–5)	0.181
Anxiety	2.85 ± 3.19	2.54 ± 3.11	2 (0–4)	2 (0–4)	0.045
Doctors (n = 58)					
Depression	2.71 ± 4.18	3.60 ± 4.50	1 (0–3)	3 (0–5)	0.018
Anxiety	2.02 ± 3.0	1.78 ± 2.59	1 (0–3)	1 (0–3)	0.259
Nursing Staff (n = 183)					
Depression	3.16 ± 3.27	3.40 ± 3.72	2 (1–5)	2 (0–5)	0.815
Anxiety	3.12 ± 3.18	2.79 ± 3.23	2 (1–4)	2 (0–4)	0.086

* Wilcoxon matched-pairs signed-rank test.

**Table 4 ijerph-18-11228-t004:** The association between individual factors and each MHO (random-effects logistic regressions).

	Depression		Anxiety	
	OR (95% CI)	*p*-Value	OR (95% CI)	*p*-Value
Age group (Ref: <31 years)				
31–40 years	1.6 (0.5–4.8)	0.419	0.5 (0.2–1.6)	0.245
41 years and above	7.9 (1.1–55.6)	0.039	1.6 (0.3–9.6)	0.631
Female	1.0 (0.3–3.2)	0.970	2.3 (0.7–7.6)	0.178
Ethnicity (Ref: Chinese)				
Filipino	0.7 (0.2–2.6)	0.617	0.5 (0.1–1.7)	0.258
Others	0.8 (0.2–2.4)	0.664	1.3 (0.4–4.1)	0.703
Married	0.4 (0.1–1.1)	0.084	1.6 (0.6–4.1)	0.374
Occupation (Ref: Physician)				
Nursing staff	1.0 (0.3–3.9)	0.998	1.4 (0.3–5.5)	0.664
Number of years in occupation	0.9 (0.8–1.0)	0.019	1.0 (0.9–1.1)	0.382
Living with elderly (Ref: No)	6.3 (1.8–21.8)	0.004	7.9 (2.3–27.2)	0.001
Living with young children (Ref: No)	0.7 (0.2–2.6)	0.601	0.1 (0–0.6)	0.006
Coping Strategies_Religion (Ref: No)	0.8 (0.2–2.8)	0.753	2.7 (0.7–10.3)	0.138
Concerns about infection risk	1.2 (0.7–2.1)	0.485	1.8 (1.03–3.0)	0.039
Concerns about workload	2.0 (1.2–3.4)	0.008	1.4 (0.9–2.4)	0.155
Concerns about working environment	1.6 (0.9–2.7)	0.058	1.5 (0.9–2.4)	0.107
Perceptions about workplace support	0.5 (0.2–0.9)	0.037	1.0 (0.5–2)	0.934
Perceptions about social connectedness	0.3 (0.1–0.6)	0.001	0.6 (0.3–1.2)	0.173

## Data Availability

The individual datasets collected and analysed will not be publicly made available due to privacy and confidentiality reasons. Data presented in this study is available upon request from the corresponding author.

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
