# Peer review of "A One-Year Longitudinal Study: Changes in Depression and Anxiety in Frontline Emergency Department Healthcare Workers in the COVID-19 Pandemic"

_ijerph, 2021, doi:10.3390/ijerph182111228_

Round 1
Reviewer 1 Report
This manuscript aims to measure changes in depression and anxiety among frontline healthcare workers during the pandemic. While I agree that there are, unfortunately, few longitudinal and well-designed studies that measure changes, and rather hundreds of cross-sectional and lower quality studies that cannot adequately answer this research question, this study has some serious methodological and analytical issues that must be addressed.
First, the two cohorts queried at two time periods were not the same. This severely limits the authors' ability to claim that they measured changes. Sentences such as "Our study found 1) a significant improvement in anxiety scores amongst all HCWs, 215 and 2) a significant worsening in depression scores amongst doctors over a 1-year period" need to be modified.
Second, almost all depression and anxiety scores displayed in Table 3 are non-normally distributed - SDs are clearly larger than mean values. Mean differences cannot be calculated from non-normally distributed data, and yet the authors do so in the third column of that table. The authors also calculate the "score difference median" and the difference between IQRs, which is not an appropriate calculation.
Related to this, the authors need to confirm whether their regression analyses were able to account for non-parametric data. I also missed model fit and other standard statistics reported.
Third, the authors need to comment on whether they made any post-hoc adjustments for conducting so many comparisons, with a sample size of only about 200.
Fourth, it appears the authors have made their own instrument to measure "concerns regarding COVID-19". If this is an unvalidated instrument, the authors need to clearly state this. It would also be valuable to thoroughly report psychometric and measurement properties. The authors mention performing a content analysis to organize items in the questionnaire into domains. This is of course appropriate in the development of a tool, but it is essential that these domains then be tested/confirmed or other explored, e.g. through a factor analysis or structural equation model. Domain and any total scoring methods must be reported.
Author Response
RESPONSE TO REVIEWER 1
Dear Reviewer,
We thank you for providing useful comments and inputs for the manuscript. Each comment has been individually addressed. As a result, the current manuscript is a much-improved version.
Point 1: First, the two cohorts queried at two time periods were not the same. This severely limits the authors' ability to claim that the measured changes. Sentences such as "Our study found 1) a significant improvement in anxiety scores amongst all HCWs, 215 and 2) a significant worsening in depression scores amongst doctors over a 1-year period" need to be modified.
Response 1: Participants in the first survey (2020) and second survey (2021) were matched for this longitudinal study. The Methods section has been amended to reflect this clearer.
Point 2: Second, almost all depression and anxiety scores displayed in Table 3 are non-normally distributed - SDs are clearly larger than mean values. Mean differences cannot be calculated from non-normally distributed data, and yet the authors do so in the third column of that table. The authors also calculate the "score difference median" and the difference between IQRs, which is not an appropriate calculation.
Response 2: We acknowledge that Median and IQR are better summary statistics to describe the distribution. The significant p-values for anxiety in all HCWs and depression amongst doctors were derived using Wilcoxon matched-pairs signed-rank test (added to the Table). We have included the Mean scores in order to illustrate these significant changes so that it would be easier for our readers to understand.
Point 3: Related to this, the authors need to confirm whether their regression analyses were able to account for non-parametric data. I also missed model fit and other standard statistics reported.
Response 3: The regression models were chosen taking into account the skewness of the depression and anxiety scores. Instead of using the depression and anxiety scores as the outcomes, we used the binary outcomes (normal vs positive for depression or anxiety) as outcomes and conducted logistic regressions. As the regression models were used to examine the association between factors and outcomes instead of for prediction, the model fit is not a useful indicator of relationships or a model’s usefulness.
Point 4: Third, the authors need to comment on whether they made any post-hoc adjustments for conducting so many comparisons, with a sample size of only about 200.
Response 4: Table 4 shows the random-effects logistic regression results of only 2 outcomes (depression and anxiety). The association between depression and anxiety, and selected factors were assessed using random-effects logistic regression adjusted for other factors. “Adjustments” were therefore with regard to each of those selected factors which included age, gender, ethnicity, marital status, occupation, no. of years in occupation, living with elderly, living with young children, coping, religion as a coping strategy, concerns about infection risk, concerns about workload, concerns about working environment, perceptions on workplace support, and perceptions on social connectedness. As these were derived from logistic regressions instead of multiple comparisons, we did not apply any post-hoc adjustments.
Point 5: Fourth, it appears the authors have made their own instrument to measure "concerns regarding COVID-19". If this is an unvalidated instrument, the authors need to clearly state this. It would also be valuable to thoroughly report psychometric and measurement properties. The authors mention performing a content analysis to organize items in the questionnaire into domains. This is of course appropriate in the development of a tool, but it is essential that these domains then be tested/confirmed or other explored, e.g. through a factor analysis or structural equation model. Domain and any total scoring methods must be reported.
Response 5: We have clarified in the Methods section that questions in Sections D and E (‘concerns regarding COVID-19’ and ‘coping strategies’) are unvalidated. We have however included the factor analysis results for each domain in these sections in Appendix C.
Reviewer 2 Report
Introduction
In which way does the pandemic stress affect sleep? In terms of sleep duration? Latency?
Material and methods
Why only a centre has been chosen? Which technique was used to recruit participants: snowball…?
Criteria for inclusión and exclusión of the participants are missing. They must be included.
Why was that precise scale chosen to measure anxiety and depression and not the Hospital Scale of Anxiety and Depression, for example, HADS?
Results
Was the gender variable taken into consideration as a factor to be used in the regression?
This could result in a bias in the results.
Discussion
Why those living with kids are less worried? It is known that covid could have serious effects on kids.
Which hypothesis would explain these results?
Why does anxiety decrease but not depression?
Which future line sor lines of treatment are proposed? It will be interesting to include them in the discussion.
Author Response
RESPONSE TO REVIEWER 2
Dear Reviewer,
We thank you for providing useful comments and inputs for the manuscript. Each comment has been individually addressed. As a result, the current manuscript is a much-improved version.
Point 1: In which way does the pandemic stress affect sleep? In terms of sleep duration? Latency?
Response 1: We had not specifically looked at sleep as one of the domains of mental wellness in this study. This is a useful suggestion to consider in future studies.
Point 2: Why only a centre has been chosen? Which technique was used to recruit participants: snowball…?
Response 2: A single centre was chosen in this longitudinal study as we wanted to match the cohort done in the first study a year ago in the same centre. No, we did not use a snowball method of recruitment. Recruitment was open to the entire emergency department of healthcare workers with clinical roles at the same time. Participants were informed of this via an email which we then followed up by making available the paper-based survey questionnaire.
Point 3: Criteria for inclusión and exclusión of the participants are missing. They must be included.
Response 3: We have amended the manuscript to reflect these clearer by including “The inclusion criteria for this study was all HCWs from this single-centre’s ED. HCWs who had resigned or had transferred out of the department during the 1 year were excluded from the second survey. As participation was voluntary, HCWs who did not wish to participate were not included in the survey.”
Point 4: Why was that precise scale chosen to measure anxiety and depression and not the Hospital Scale of Anxiety and Depression, for example, HADS?
Response 4: We chose the DASS-21 as it was a well-established instrument to measure depression, anxiety and stress as opposed to the HADS for example which only measures 2 states of psychological distress. The DASS-21 has good validity and has been reliably used in different ethnic groups ranging from Chinese, Hindi to Spanish, which is an important consideration given our multi-ethnic study population. The DASS-21 has also been widely used in many COVID-19 related studies on mental health.
Point 5: Was the gender variable taken into consideration as a factor to be used in the regression? This could result in a bias in the results.
Response 5: Gender was included as a factor in the regression. We examined the collinearity of the included variables and did not observe any collinearity issue by including gender in the model.
Point 6: Why those living with kids are less worried? It is known that covid could have serious effects on kids. Which hypothesis would explain these results?
Response 6: We mentioned in lines 273-278 of the manuscript that we were also surprised by this result in those living with kids, and it was contrary to other studies’ findings which we had listed in the manuscript. A possible hypothesis would be at the time when we conducted the survey, the understanding was that children were less affected by COVID-19 and had far less fatalities.
Point 7: Why does anxiety decrease but not depression?
Response 7: Lines 279-288 of the manuscript contained our explanation as to why anxiety decreased in our cohort as a whole. Lines 290-326 of the manuscript explained why depression increased in doctors. Even though absolute mean scores of depression also increased in the nurses, it was not statistically significant compared to the doctors group. In summary, we think that anxiety and depression are affected by different mechanisms as detailed in these explanations in the manuscript.
Point 8: Which future line or lines of treatment are proposed? It will be interesting to include them in the discussion.
Response 8: We have included the following in the manuscript: “The implications of this longitudinal study are firstly, we need to recognise that effects of psychological distress inflicted by a pandemic are long-standing. From an occupational health standpoint, workplace mental health interventions cannot be sporadic, but should be regular and sustained. HCWs should continue to receive updated information about infection protocols and risk management, as well as redistribution of workload from the fever area for those most affected.
Secondly, this longitudinal study builds on the understanding that certain factors associated with depression and anxiety are non-modifiable, such as those older than 40 years of age and those living with elderly. Gaining a better understanding of how these groups connect to fellow beings, whether through social media or lack thereof, will allow us to develop mental health programs targeted towards them. The results of the survey will also be shared with department heads and nursing leads so that they can better identify the at-risk groups to attend hospital-wide wellness initiatives.”
Reviewer 3 Report
Reviewer comments for authors
Thank you for the opportunity to review the manuscript titled “A One-Year Longitudinal study: Changes in Depression and Anxiety in Frontline Emergency Department Healthcare Workers in the COVID-19 Pandemic.” The findings of this study will help in designing targeted interventions to promote psychological well-being of healthcare professionals. Despite this contribution, I believe there are opportunities to further strengthen this manuscript. I will share these in the order that they appear.
- Title
Title is descriptive and reflective of the manuscript’s content.
- Abstract
- A new sentence does not start with a numeric form of a figure. Please see Line # 14, it is recommended to start with the word form i.e. Two hundred and forty-one. Please indicate how these participants were recruited?
- Abstract needs extensive English editing. There are several run-on and confusing sentence structures used. Issues with capitalizations too.
- In the keywords, there is no need to assign numbering.
- Introduction
- The introduction section not back up by sufficient literature review. Recent references can be added to strengthen the argument of psychosocial impact and its negative consequences among frontline healthcare professionals. The suggested reference can be worth adding.
https://www.mdpi.com/1660-4601/17/23/9096
- Methods
- Which platform was used for an online survey?
- Please describe the sampling strategy in depth.
- How multiple responses from the same participants were prevented?
- Did authors conduct a priori power analysis? If yes, then provide the details. If no, please provide the appropriate justification.
- What did authors do to control the type 1 error in multiple comparisons of Chi/Fisher?
- Why authors limited their analysis to the levels of anxiety and depression. They could have analyzed the subscales scores of DASS-21. For example:
- Results
- Table 1, why p values for gender and ethnicity are missing?
- The interpretation of odds ratio should be in terms of “odds” rather than “likely”.
- Discussion
- The discussion section is weak and is not back up by sufficient literature review. Please use the aforementioned reference to back up the discussion.
- Please add more on the potential value of this study. For example, post-traumatic growth.
- Please compare your findings with other population groups in the context of psychological impact of COVID-19. The reference suggested below can be worth adding:
https://pubmed.ncbi.nlm.nih.gov/33671363/
Author Response
RESPONSE TO REVIEWER 3
Dear Reviewer,
We thank you for providing useful comments and inputs for the manuscript. Each comment has been individually addressed. As a result, the current manuscript is a much-improved version.
Point 1: Title. Title is descriptive and reflective of the manuscript’s content.
Point 2: Abstract. A new sentence does not start with a numeric form of a figure. Please see Line # 14, it is recommended to start with the word form i.e. Two hundred and forty-one.
Response 2: This sentence has been re-written.
Point 3: Please indicate how these participants were recruited?
Response 3: We have edited the abstract to briefly describe the recruitment process. However, due to the word limit, the detailed sampling process is expounded on in the Methods section. The following has been added to the Abstract: “In this longitudinal single-centre study in Singapore, all ED HCWs were prospectively recruited face-to-face. Paper-based surveys were administered in June 2020 and June 2021.”
Point 4: Abstract needs extensive English editing. There are several run-on and confusing sentence structures used. Issues with capitalizations too. In the keywords, there is no need to assign numbering.
Response 4: We have taken this feedback and extensively revised the abstract for grammar and readers’ understanding.
Point 5: Introduction. The introduction section not back up by sufficient literature review. Recent references can be added to strengthen the argument of psychosocial impact and its negative consequences among frontline healthcare professionals. The suggested reference can be worth adding. https://www.mdpi.com/1660-4601/17/23/9096
Response 5: Introduction has been amended to include the above reference.
Point 6: Methods. Which platform was used for an online survey?
Response 6: Surveys were paper-based. Lines 113-114: “Paper-based consent forms and survey questionnaires were handed out to the eligible ED HCWs during roll calls.”
Point 7: Please describe the sampling strategy in depth.
Response 7: As this is a population survey of all eligible Emergency Department Healthcare workers, we did not apply a sampling strategy.
Point 8: How multiple responses from the same participants were prevented?
Response 8: We have amended the manuscript to include this: “The ED register list was used to ensure that surveys were only handed out to each participant once”.
Point 9: Did authors conduct a priori power analysis? If yes, then provide the details. If no, please provide the appropriate justification.
Response 9: We did not conduct a power analysis as our survey was a population-based survey. The overall response rate for those that meet inclusion/exclusion criteria is 93.4%.
Point 10: What did authors do to control the type 1 error in multiple comparisons of Chi/Fisher?
Response 10: The Chi-square tests and Fisher’s exact tests were used to:
1) examine whether there was difference in distribution and status of depression and anxiety;
2) examine the univariate association between different identified factors and outcomes for selecting factors to be included in the multiple regression model instead of comparison across multiple groups of participants or making hypotheses to draw any conclusions. As such, we did not perceive a need to control any potential error related to multiple comparisons.
Point 11: Why authors limited their analysis to the levels of anxiety and depression. They could have analyzed the subscales scores of DASS-21.
Response 11: The emphasis of this study was on anxiety and depression only. Hence, we did not report on the stress subscale as it would dilute the focus of the paper.
Point 12: Results. Table 1, why p values for gender and ethnicity are missing?
Response 12: As participants were the same, there was no change in gender and ethnicity. The p value in Table 1 reflects any significant changes between the 2020 and 2021 results.
Point 13: The interpretation of odds ratio should be in terms of “odds” rather than “likely”.
Response 13: We agree. This has been amended.
Point 14: Discussion. The discussion section is weak and is not back up by sufficient literature review. Please use the aforementioned reference to back up the discussion. Please add more on the potential value of this study. For example, post-traumatic growth.
Response 14: The following have been added to the manuscript:
“It has also been widely reported that nurses experience primary and secondary traumatic stress, compassion fatigue and burnout during this COVID-19 pandemic, such as in this Chinese study [33]. However, another Spanish study [34] reported that physicians actually have higher compassion fatigue and burnout scores compared to nurses, demonstrating that both parties are equally vulnerable to emotional desensitization.”
“The findings in our study emphasize the importance and need to develop psychological interventions to promote post-traumatic growth amongst our frontline ED HCWs.”
“The implications of this longitudinal study are firstly, we need to recognise that effects of psychological distress inflicted by a pandemic are long-standing. From an occupational health standpoint, workplace mental health interventions cannot be sporadic, but should be regular and sustained. HCWs should continue to receive updated information about infection protocols and risk management, as well as redistribution of workload into the fever area for those most affected.
Secondly, this longitudinal study builds on the understanding that certain factors associated with depression and anxiety are non-modifiable, such as those older than 40 years of age and those living with elderly. Gaining a better understanding of how these groups connect to fellow beings, whether through social media or lack thereof, will allow us to develop mental health programs targeted towards them. The results of the survey will also be shared with department heads and nursing leads so that they can better identify the at-risk groups to attend hospital-wide wellness initiatives.”
Point 15: Please compare your findings with other population groups in the context of psychological impact of COVID-19. The reference suggested below can be worth adding: https://pubmed.ncbi.nlm.nih.gov/33671363/
Response 15: We have included the above reference in our manuscript. The following has been added: “ Interestingly, another population group that has demonstrated similar poor MHOs as HCWs during this COVID-19 pandemic are college students. A large meta-analysis [20] consisting of 27 studies with 90,879 college students indicated a prevalence of 31.2% in depression and 39.4% in anxiety, with females having higher depression and anxiety than males.”
Reviewer 4 Report
Dear authors,
In this manuscript, Th'ng and colleagues describe changes in the emotional reaction of frontline HCW over the course of one year. It is an interesting study, and it should have implications for improving mental wellbeing of HCW. However, the manuscript needs further modifications to be ready for publication. The comments below may help make it better.
Generally, do not begin a sentence with a number such as “241 HCWs were included”, which you have written in the abstract.
Describe mean age, age range, % of males or females, location of the study, composition of the sample (e.g., % of doctors and nurses) in the abstract. So, you can rewrite that sentence as follows “The study included 241 HCWs (mean age …., age range, …%males, …%doctors) working at …in [country name].
Why are you using both the mean and median to describe your statistics? The median values “Median: 2020:2(0-4) vs. 2021:2(0-4)” indicate no change from 2020 to 2021, which is contradictory to the result that you are reporting.
I believe that you need to revise the whole manuscript linguistically; for example, “concerns in Infection Risk” should be “concerns about infection risk”. The same goes for “with greater concerns in Workload”. Note the change in the preposition. “Infection Risk”, “Workload”, “Social Connectedness”, and the like should not come in capital letters.
Remove unnecessary phrases e.g., if living with the elderly would correlate with high levels of depression and anxiety, it is better to write this result in a single sentence “HCWs living with elderly ……... Amongst 241 HCWs, those ≥41 years, living with elderly and with greater concerns in Workload had higher odds of depression.”
Since you are addressing change over time in the emotional reaction “within the context of fighting a prolonged pandemic, line 64”, it would be very beneficial to elaborate a bit in the Introduction on the concepts of coping, resilience, adaptation during COVID-19, and how they relate to mental wellbeing during the pandemic.
In the Introduction, exploring the relation between burnout in clinical setting and mental dysfunction would reasonable to pave the way for workload as a probable cause of depression in your results. The same may also hold for social support and mental wellbeing.
It is not clear in line 292-294 if you are discussing your present results or those of previous studies—please revise.
Both anxiety and depression were higher among HCW living with the elderly. You need to discuss why—extra workload or concerns about transmitting infection to their dear seniors. Young children can also be troublesome because of stay-at-home-orders, which is documented to increase parental distress. So, maybe the sociodemographic characteristics have a role in these relations. I expect that those living with the elderly are more likely to be single while those who have children may receive support from their spouses. Exploring the difference in social connectedness between the two groups will clarify the mystery.
In your last year survey, nurses had higher levels of depression and this year only doctors have higher levels. Is there a chance that nurses’ more close contact with critical patients “than doctors” caused emotional desensitization, which resulted in the reduction of their depression? Please discuss within the light of the available literature.
To have a practical value, you need to note the implications added by this study in the conclusion on the abstract and in the Discussion as well.
As for the limitations, I think you need to note concerns about the psychometric properties of the measures of coping strategies and social connectedness—my understanding is that you did not use well-calibrated specific measures and that you developed a set of questions within your group. I agree that the DASS-21 has been extensively; however, its psychometrics are not-established. The DASS-21 has been extensively used during the COVID-19 pandemic as a measure of depression and anxiety, but numerous studies [“doi: 10.1186/s13011-019-0226-1” and “doi:10.3390/ijerph181910142 ”, please cite] highlight its inability to differentiate between depression and anxiety. So, the self-report nature of depression and anxiety rather than clinician-facilitated assessment can really be a major limitation that should be mentioned.
Hoping these comments can be helpful and wishing you good luck.
Author Response
RESPONSE TO REVIEWER 4
Dear Reviewer,
We thank you for providing useful comments and inputs for the manuscript. Each comment has been individually addressed. As a result, the current manuscript is a much-improved version.
Point 1: Generally, do not begin a sentence with a number such as “241 HCWs were included”, which you have written in the abstract.
Response 1: This sentence has been re-written.
Point 2: Describe mean age, age range, % of males or females, location of the study, composition of the sample (e.g., % of doctors and nurses) in the abstract. So, you can rewrite that sentence as follows “The study included 241 HCWs (mean age …., age range, …%males, …%doctors) working at …in [country name].
Response 2: We have included that the study was carried out in Singapore. We have left out further demographics in the Abstract due to the word limit.
Point 3: Why are you using both the mean and median to describe your statistics? The median values “Median: 2020:2(0-4) vs. 2021:2(0-4)” indicate no change from 2020 to 2021, which is contradictory to the result that you are reporting.
Response 3: We acknowledge that Median and IQR are better summary statistics to describe the distribution. The significant p-values for anxiety in all HCWs and depression amongst doctors were derived using Wilcoxon matched-pairs signed-rank test (added to the Table). We have included the Mean scores in order to illustrate these significant changes so that it would be easier for our readers to understand.
Point 4: I believe that you need to revise the whole manuscript linguistically; for example, “concerns in Infection Risk” should be “concerns about infection risk”. The same goes for “with greater concerns in Workload”. Note the change in the preposition. “Infection Risk”, “Workload”, “Social Connectedness”, and the like should not come in capital letters.
Response 4: These have been amended accordingly.
Point 5: Remove unnecessary phrases e.g., if living with the elderly would correlate with high levels of depression and anxiety, it is better to write this result in a single sentence “HCWs living with elderly ……... Amongst 241 HCWs, those ≥41 years, living with elderly and with greater concerns in Workload had higher odds of depression.”
Response 5: These have been amended accordingly.
Point 6: Since you are addressing change over time in the emotional reaction “within the context of fighting a prolonged pandemic, line 64”, it would be very beneficial to elaborate a bit in the Introduction on the concepts of coping, resilience, adaptation during COVID-19, and how they relate to mental wellbeing during the pandemic.
Response 6: This has been added to the Introduction.
Point 7: In the Introduction, exploring the relation between burnout in clinical setting and mental dysfunction would reasonable to pave the way for workload as a probable cause of depression in your results. The same may also hold for social support and mental wellbeing.
Response 7: The relationship between burnout and depression was explained in lines 303-310. We had included it in the subheading of “Increased depression in doctors” as opposed to the introduction as the worsening depression scores were only significant in the doctor group and not as a whole. In the univariate analysis as detailed in Appendix D, workload was found to be associated with higher odds of depression, while workplace support and social connectedness were found to be associated with lower odds of depression.
Point 8: It is not clear in line 292-294 if you are discussing your present results or those of previous studies—please revise.
Response 8: We have changed the wording to “In our current study” instead of “In our cohort of ED HCWs”.
Point 9: Both anxiety and depression were higher among HCW living with the elderly. You need to discuss why—extra workload or concerns about transmitting infection to their dear seniors. Young children can also be troublesome because of stay-at-home-orders, which is documented to increase parental distress. So, maybe the sociodemographic characteristics have a role in these relations. I expect that those living with the elderly are more likely to be single while those who have children may receive support from their spouses. Exploring the difference in social connectedness between the two groups will clarify the mystery.
Response 9: We had tried to explain this phenomenon in our first study (reference 12 in the manuscript). It is a combination of the time and energy demanded from caregiving, as well as the fear of infecting the elderly who are the most vulnerable group in this pandemic. Our hypothesis as to why those with young children are not as affected is because at the time when we did this survey, morbidity and mortality was low amongst children.
The literature is not robust on whether those living with elderly are more likely to be single. As such, we conducted further analysis based on our demographic data to see if there is any relationship between marital status and living with elderly. Based on our results, those living with elderly had slightly higher proportion of married individuals (53%), demonstrating no significant relationship between the two. This could be cultural related as large extended families continue to be prevalent in Asian societies, hence the likelihood of an elderly living with their married children is higher.
In addition, we also further analysed the difference in social connectedness between those who are single and those who are married (and hence presumably have spousal support). Our analysis also revealed no significant difference in social connectedness between the 2 groups.
Point 10: In your last year survey, nurses had higher levels of depression and this year only doctors have higher levels. Is there a chance that nurses’ more close contact with critical patients “than doctors” caused emotional desensitization, which resulted in the reduction of their depression? Please discuss within the light of the available literature.
Response 10: We attempted to explain why doctors may be more prone to depression in Lines 298-326. We have also added the following: “It has also been widely reported that nurses experience primary and secondary traumatic stress, compassion fatigue and burnout during this COVID-19 pandemic, such as in this Chinese study [34]. However, another Spanish study [35] reported that physicians actually have higher compassion fatigue and burnout scores compared to nurses, demonstrating that both parties are equally vulnerable to emotional desensitization.”
Point 11: To have a practical value, you need to note the implications added by this study in the conclusion on the abstract and in the Discussion as well.
Response 11: We have added this to the last paragraph of the Conclusion:
“The implications of this longitudinal study are firstly, we need to recognise that effects of psychological distress inflicted by a pandemic are long-standing. From an occupational health standpoint, workplace mental health interventions cannot be sporadic, but should be regular and sustained. HCWs should continue to receive updated information about infection protocols and risk management, as well as redistribution of workload into the fever area for those most affected.
Secondly, this longitudinal study builds on the understanding that certain factors associated with depression and anxiety are non-modifiable, such as those older than 40 years of age and those living with elderly. Gaining a better understanding of how these groups connect to fellow beings, whether through social media or lack thereof, will allow us to develop mental health programs targeted towards them. The results of the survey will also be shared with department heads and nursing leads so that they can better identify the at-risk groups to attend hospital-wide wellness initiatives.”
Point 12: As for the limitations, I think you need to note concerns about the psychometric properties of the measures of coping strategies and social connectedness—my understanding is that you did not use well-calibrated specific measures and that you developed a set of questions within your group. I agree that the DASS-21 has been extensively; however, its psychometrics are not-established. The DASS-21 has been extensively used during the COVID-19 pandemic as a measure of depression and anxiety, but numerous studies [“doi: 10.1186/s13011-019-0226-1” and “doi:10.3390/ijerph181910142 ”, please cite] highlight its inability to differentiate between depression and anxiety. So, the self-report nature of depression and anxiety rather than clinician-facilitated assessment can really be a major limitation that should be mentioned.
Response 12: Thank you for raising this. We agree that the self-report nature of DASS-21 scoring is a major limitation and will add this to the manuscript. In addition, we acknowledge that we had included questions on concerns and coping strategies in Sections D and E of the questionnaire that were not part of a validated instrument and we had addressed this in the section of “Outcome Measures” in the manuscript.
Round 2
Reviewer 1 Report
- My major concern in the first round was that the study design was repeated cross-sectional (two different groups of participants were measured at two time points), and yet the authors consistently described their "longitudinal" results as showing changes, implying that they measured between-person differences. The authors have now changed their descriptions of their methods to say that the same participants contributed data at both time points. At the same time, the authors also write that participants' answers were anonymized: "Responses from the first and second waves of survey were anonymised, but they were matched based on the last four digits of the HCW’s handphone 117 number, gender, ethnicity, and profession. In total, 241 participants were matched and 118 their responses to both the first and second waves of survey were included for data analysis." Something is wrong here about the methods: in the first version of the paper, the participants were different at two time points; in this version, the participants were the same and their data was anonymized, yet this anonymized data was identifiable, meaning it was not anonymized. This is either a study design problem or an ethical problem (as the authors appear to be being disingenuous in their descriptions).
- In addition, none of my earlier concerns appear to have been addressed. Here I am copying in my previous concerns: " Second, almost all depression and anxiety scores displayed in Table 3 are nonparametric - SDs are clearly larger than mean values. Mean differences cannot be calculated from non-normally distributed data, and yet the authors do so in the third column of that table. The authors also calculate the "score difference median" and the difference between IQRs, which is not an appropriate calculation.
- Related to this, the authors need to confirm whether their regression analyses were able to account for non-parametric data. I also missed model fit and other standard statistics reported.
- Third, the authors need to comment on whether they made any post-hoc adjustments for conducting so many comparisons, with a sample size of only about 200.
- Fourth, it appears the authors have made their own instrument to measure "concerns regarding COVID-19". If this is an unvalidated instrument, the authors need to clearly state this. It would also be valuable to thoroughly report psychometric and measurement properties. The authors mention performing a content analysis to organize items in the questionnaire into domains. This is of course appropriate in the development of a tool, but it is essential that these domains then be tested/confirmed or other explored, e.g. through a factor analysis or structural equation model."
Reviewer 4 Report
Dear authors,
Thank you for your hard effort. I have no no further comments.
Best regards
Author Response
Thank you very much.
Best reagrds.